# Epigenetic Changes in Prion and Prion-like Neurodegenerative Diseases: Recent Advances, Potential as Biomarkers, and Future Perspectives

**DOI:** 10.3390/ijms232012609

**Published:** 2022-10-20

**Authors:** Adelaida Hernaiz, Janne Markus Toivonen, Rosa Bolea, Inmaculada Martín-Burriel

**Affiliations:** 1Laboratorio de Genética Bioquímica (LAGENBIO), Facultad de Veterinaria, Universidad de Zaragoza, IA2, IIS Aragón, 50013 Zaragoza, Spain; 2Centro de Investigación Biomédica en Red de Enfermedades Neurodegenerativas (CIBERNED), Instituto Carlos III, 28029 Madrid, Spain; 3Centro de Encefalopatías y Enfermedades Transmisibles Emergentes (CEETE), Facultad de Veterinaria, Universidad de Zaragoza, IA2, IIS Aragón, 50013 Zaragoza, Spain

**Keywords:** epigenetics, DNA methylation, histone modifications, microRNA, prion diseases, prion-like diseases

## Abstract

Prion diseases are transmissible spongiform encephalopathies (TSEs) caused by a conformational conversion of the native cellular prion protein (PrP^C^) to an abnormal, infectious isoform called PrP^Sc^. Amyotrophic lateral sclerosis, Alzheimer’s, Parkinson’s, and Huntington’s diseases are also known as prion-like diseases because they share common features with prion diseases, including protein misfolding and aggregation, as well as the spread of these misfolded proteins into different brain regions. Increasing evidence proposes the involvement of epigenetic mechanisms, namely DNA methylation, post-translational modifications of histones, and microRNA-mediated post-transcriptional gene regulation in the pathogenesis of prion-like diseases. Little is known about the role of epigenetic modifications in prion diseases, but recent findings also point to a potential regulatory role of epigenetic mechanisms in the pathology of these diseases. This review highlights recent findings on epigenetic modifications in TSEs and prion-like diseases and discusses the potential role of such mechanisms in disease pathology and their use as potential biomarkers.

## 1. Introduction

Epigenetics is the study of heritable changes in gene activity or function that is not associated with any change in the DNA sequence itself [1]. Three major epigenetic mechanisms have been described: DNA methylation and histone modifications essentially alter the structure of the nearby chromatin of a particular gene or region, whereas non-coding RNAs, especially microRNAs (miRNAs), may regulate expression without the necessity for the physical vicinity for their target genes.

DNA methylation involves the covalent transfer of a methyl group to the C-5 position of the nucleobase cytosine to form 5-methylcytosine. In mammals, DNA methylation may occur at cytosines in any context of the genome [1] and can participate in the regulation of gene expression. Histones are structural proteins that form DNA nucleosomes and are frequently subjected to covalent post-translational modifications. They are involved in the expression and repression of target genes generally via chromatin modification [2]. The most studied histone modifications are acetylation, methylation, and phosphorylation, but also other modifications exist, including citrullination, ubiquitination, ADP-ribosylation, deamination, and proline isomerization [2,3].

MicroRNAs are small non-coding RNAs (21–23 nucleotides in length) that may regulate gene expression at transcriptional, posttranscriptional, or translational levels [4]. Although miRNAs may induce gene expression in some cases by their interaction with gene promoters [5], miRNA-based regulation is thought to rely largely on the repression of target mRNA translation or its sequence-dependent degradation. Several miRNAs display a cell- or tissue-specific expression profile, while others are widely expressed. They can be found in blood and other biofluids as free (mainly protein-associated) complexes or can be contained by extracellular vesicles. Moreover, circulating miRNAs can be detected and measured by highly sensitive and specific methods (e.g., quantitative PCR and next-generation sequencing). All these characteristics have facilitated studies of miRNAs as potential biomarker molecules [6].

The mentioned epigenetic mechanisms are involved in several aspects of brain development as well as in normal aging [7,8]. DNA methylation and histone acetylation are essential in memory acquirement, learning, and acquisition of long-term memories [8,9,10], and miRNAs are crucial for the formation and maturation of synapses and for dendritogenesis in early brain development [11,12,13]. Considering the central role of epigenetics in neural plasticity, it is not surprising that dysregulation of these processes has been associated with different neurodegenerative diseases, either by mediating interactions between genetic and environmental risk factors or by directly interacting with disease-specific pathological factors [7,8]. The neurodegenerative disorders where epigenetics seem to have a key role include Alzheimer’s (AD), Parkinson’s (PD), and Huntington’s (HD) diseases, amyotrophic lateral sclerosis (ALS), prion diseases, stroke, and global ischemia [7,8]. The first four are also known as prion-like diseases because they are proteinopathies that share common pathogenic mechanisms with prion diseases, including the accumulation of misfolded proteins in the central nervous system (CNS) [14,15].

Prion diseases, also referred to as transmissible spongiform encephalopathies (TSEs), are a group of neurodegenerative disorders affecting humans and other animals [16] and are caused by a conformational conversion of the cellular prion protein (PrP^C^) to an infectious isoform, partially resistant to proteases and prone to form aggregates called PrP^Sc^ [17].

Research in yeast has revealed that certain proteins that adopt prion conformation, such as URE3 or PSI^+^, can be considered as an epigenetic mechanism that can be inherited through mitosis and meiosis. These self-templating conformations of prion proteins interact with nucleic acids (DNA and RNA) and can regulate gene expression through the modification of chromatin remodeling, nucleic acid translation, and replication, providing beneficial phenotypes in stressful conditions [18]. If prions in yeast possess these characteristics, it could be expected that alteration of the prion protein and other proteins with similar characteristics in mammals will also affect epigenetic mechanisms. Increasing evidence suggests involvement of epigenetic mechanisms in the pathogenesis of prion-like diseases and other neurodegenerative disorders. However, current knowledge about the association of epigenetics with prion diseases is scarce.

Given that some of the affected mechanisms may be shared with TSEs and prion-like diseases, this review summarizes the latest findings on epigenetic modifications in prion-like diseases and reports the recently discovered roles of epigenetic mechanisms in TSEs.

## 2. Epigenetic Changes in Prion-Like Diseases

### 2.1. DNA Methylation

#### 2.1.1. DNA Methylation Profiles

Altered DNA methylation patterns have been associated with the neuropathology of prion-like diseases. Several studies have identified sets of genes containing differentially methylated regions (DMRs) and differentially methylated positions (DMPs) between patients and healthy controls, primarily in peripheral blood [19,20,21,22,23,24,25,26] and brain tissue [27,28,29,30,31,32,33,34,35] and to a lesser extent in saliva [23] and using in vitro models [36,37]. Table 1 summarizes global and gene-specific DNA methylation changes reported in these diseases.

When comparing the DMRs and DMPs identified in each prion-like disease (Appendix A) using the InteractiVenn software [38], no common DMRs and DMPs were found between all four diseases, but some regions and positions matched between two or three of these diseases (Figure 1). Regarding the DMRs, the disease pairs that shared the highest number of common regions were first PD and HD, second PD and ALS, and third PD and AD (Figure 1a). On the other hand, PD and AD are the only diseases that share a relatively high number of DMPs (Figure 1b). Although these comparisons between the DMRs and DMPs identified in prion-like diseases allow an overview of common differentially methylated genes shared between some of these diseases, there are limitations that could affect the number of regions and positions identified in each disease and must be taken into account: (1) the compared studies were performed in different tissues and body fluids which could lead to different methylation profiles inherent to each tissue; (2) the number of studies developed in each disease is different, being AD and PD the most studied ones; and (3) the methodology used in the different works varies. Most studies were performed using the 450 K methylation array, and whole genome bisulfite sequencing was used only in one study of PD [37]. This methodology detects DMPs in non-annotated sequences such as lincRNAs, pseudogenes, and antisense or unknown miRNAs, which explains in part why the number of DMPs and DMRs in PD is substantially higher than in other diseases.

Studies on the DNA methylation process have concentrated on different areas of the CNS depending on their relevance with the pathogenesis of each disease. Epigenome-wide studies on AD patients have identified DMRs in the superior temporal gyrus, a region that displays a marked gene dysregulation in AD [27,28]. Most of these DMRs were found to be in gene promoters and associated with AD pathology. Significant DNA methylation changes are also found in PD-affected brain areas, namely the dorsal motor nucleus of the vagus, substantia nigra, and cingulate gyrus [32]. In HD, DNA methylation studies performed so far have shown disparate results. A genome-wide DNA methylation profile of cortex tissues from HD patients suggests that DNA methylation may have a minimal association with HD status but could be correlated with the age of disease onset and contribute to the tissue-specific expression patterns of huntingtin (*HTT*) [39]. In contrast, using the same methodology but analyzing multiple CNS regions, another study identified 11 co-methylation modules associated with HD status in cortical regions and observed an accelerated epigenetic age in specific brain regions (frontal lobe, parietal lobe, and cingulate gyrus) of HD patients [35], measured by combining DNA methylation levels of known CpGs [40].

Genome-wide studies have detected DNA modifications in specific genes that have been analyzed in detail to elucidate their role in neurodegeneration. Epigenetic control of enhancers seems to be involved in AD as loss of CpH (most frequently at CpA sites) methylation of enhancers, which is normal in aging neurons but is accelerated and occurs early in AD neurons. This modification seems to prompt the reactivation of cell cycle and neurogenesis pathways and, in the case of the enhancer of *DSCAML1* that codes for Down Syndrome cell adhesion molecule-like protein 1, precedes the onset of neurofibrillary tangle pathology [41]. Hypomethylation of *DSCAML1* gene enhancer is associated with an upregulation of *BACE1* (beta-secretase 1) transcripts and an increase in amyloid plaques, neurofibrillary tangles, and cognitive decline [41]. These results support an early involvement of epigenetic changes in AD.

In the same disease, a protective role has been suggested for *PM20D1* (peptidase M20 domain containing 1). DNA methylation and RNA expression of this gene are associated with AD, and its overexpression in cells is neuroprotective against AD stressors [42].

Methylation changes in other genes have been related to prion-like disease pathogenesis, although its role is still unknown. In PD, patients show hypomethylation of the promoter region of *SNCA*, the gene encoding α-synuclein protein [43]. Moreover, several ALS-related genes (*DENND11*, *COL15A1*, *TARDBP*, *RANGAP1*, and *IGHMBP2*) and DNA repair genes (*OGG1*, *APEX1*, *PNKP*, and *APTX*) are differentially methylated in ALS patients [44,45,46]. Differential DNA methylation has also been observed in specific genes in HD, namely *HES4*, a transcription factor involved in neural stem cell regeneration, and *BDNF*, encoding brain-derived neurotrophic factor [47,48]. *HES4* promoter hypermethylation is associated with reduced expression of the gene and with striatal degeneration and age of onset of HD patients. Consistently, inhibition of *HES4* by shRNA increases mutant HTT aggregates in a cell model of the disease. *BDNF* promoter hypermethylation in the blood of HD patients did not correlate with motor or cognitive status but may represent a biomarker for HD-associated psychiatric symptoms.

Most of the reported studies reveal that changes in DNA methylation can be associated with pathogenic processes of these neurodegenerative diseases, but there does not appear to be a universal mechanism relating DNA methylation and protein aggregates. Studies performed in human patients are normally carried out at the late stages of the disease when it is difficult to differentiate if changes in DNA methylation are a cause or a consequence of neurodegeneration. Analysis of this epigenetic process in animal or cellular models could possibly allow us to elucidate this question.

A limitation of most of these studies is the technology used for the determination of DNA methylation. Methylation arrays or NGS-sequencing-based methods did not differentiate between methyl Cytosines (mC) and their oxidized product hydroxymethyl Cytosines (hmC), which has regulatory functions and could be an epigenetic mark in its own right [49]. Antibody-based techniques have already been used in these diseases to specifically detect 5mC and 5hmC methylation forms [50,51]. However, these techniques cannot identify specific genes of interest. Third-generation sequencers [52] could help elucidate which epigenetic mark is involved in a gene-specific regulation.

#### 2.1.2. Biomarkers Based on DNA Methylation

DNA methylation as a potential biomarker has been investigated in easily accessible tissues at both genomic and gene-specific levels. Both sporadic (SALS) and familial ALS (FALS) patients show increased global 5-methylcytosine levels in blood DNA [53]. At the gene level, hypermethylation of the promoter for *C9orf72*, a gene responsible for the majority of the FALS cases (as well as for those of frontotemporal dementia), correlates with its reduced mRNA expression levels in a clinical cohort of *C9orf72* pathological expansion carriers [54]. A significant association between DNA methylation age-acceleration, disease duration, and age of onset has also been observed in *C9orf72* carriers [55]. The second most common cause of FALS is mutations in superoxide dismutase 1 (*SOD1*), with nearly 200 mutations described, some of which show incomplete penetrance and great phenotypic variability. ALS patients carrying not fully penetrant *SOD1* mutations display an increase in global DNA methylation, and DNA methylation levels correlate positively with disease duration [56]. However, since the promoters of four major ALS genes (*C9orf72*, *SOD1*, *TARDBP*, and *FUS)* were not methylated in the study subjects, it was concluded that the increased methylation is likely to occur in other gene regions. In this study, no repeat expansion was observed in *C9orf72*, which supports the previous finding [51] that gene-specific methylation in the *C9orf72* locus is dependent on this pathological feature.

In contrast, DNA hypomethylation seems to be a key mark in PD patients, finding several hypomethylated and upregulated genes in blood and saliva samples associated with systemic immune response pathways and mitochondrial dysfunction [21,23]. In addition, in leukocytes from PD patients, hypomethylation of *NPAS2* (neuronal PAS domain protein 2) [57] and *DRD2* (dopamine receptor D2) [57] has been proposed as a novel biomarker for PD.

In blood from AD patients, *B3GALT4* (beta-1,3-galactosyltransferase 4), a gene associated with AD onset and progression, and *PTGR3* (prostaglandin reductase 3), associated with AD risk, are hypomethylated and correlate with memory performance and cerebrospinal fluid (CSF) levels of Aβ and tau [58]. Moreover, hypomethylation of *BIN1* (bridging integrator 1), a gene associated with AD pathogenesis, and hypermethylation of estrogen receptor α (*ESR1*) gene promoter, which is related to impaired cognitive function and quality of life of AD patients, have been also reported [59,60]. Another potential biomarker of this disease could be the hypomethylation in *TOMM40* (translocase of outer mitochondrial membrane 40) and *APOE* (apolipoprotein E) gene promoters observed in the hippocampus, cerebellum, and peripheral blood of AD patients, which correlated with increasing *APOE* and decreasing *TOMM40* expression [61]. Besides these promising results, very few have been subjected to clinical trials. Only methylation levels on *COASY* (Coenzyme A synthase) and *SPINT1* (Serine peptidase inhibitor Kunitz type 1) promoter regions have been considered convenient and useful biomarkers for AD [62].

In HD, DNA methylation status in peripheral blood does not seem to be affected as, after a microarray study of blood samples, no distinctive patterns were observed in the covered CpG sites and their associated genes [25].

#### 2.1.3. In Vitro Studies

Cellular models are particularly useful for analyzing the effect of DNA methylation and investigating the role of methylation in candidate genes and potential treatments. DNA methylation changes in prion-like diseases have been investigated in vitro using cell models. A whole-genome bisulfite sequencing study using induced pluripotent stem cell (iPSC)-derived dopaminergic neurons from sporadic PD and monogenic *LRRK2* (Leucine-rich repeat kinase 2)-associated PD patients revealed global DNA hypermethylation associated with disease [37], conversely to the hypomethylation observed in blood and saliva. In ALS, a study that evaluated the methylation status of human embryonic stem cells (hESCs) and iPSCs both carrying the *C9orf72* mutation showed that hESCs were completely unmethylated at the *C9orf72* repeats, whereas iPSCs were hypermethylated. This methylation status remained unchanged after the differentiation of hESCs and iPSCs into neural precursors. The hypermethylation observed in the *C9orf72* repeats of iPSCs was proposed as a possible neuroprotective mechanism attenuating the accumulation of potentially toxic repeat-containing mRNAs in neurons, given that hESCs presented a more severe phenotype than iPSCs [63].

Hypomethylation of the *SNCA* gene has been described in early onset PD patients [43], and lowered methylation status is likely to increase *SNCA* expression, contributing to the accumulation of α-synuclein in this disease. With the purpose of maintaining normal physiological levels of α-synuclein, Kantor et al. [64] experimentally methylated the *SNCA* gene using a system based on CRISPR-deactivated Cas9 fused with the catalytic domain of *DNMT3A* (DNA methyltransferase 3 alpha). This technique was applied to iPSC-derived dopaminergic neurons from a PD patient resulting in hypermethylation of the *SNCA* gene, downregulation of the SNCA expression, and a reversion of disease-related phenotypic perturbations [65]. The same hypermethylation-based approach has been recently used to manipulate the levels of amyloid-beta (Aβ) precursor (APP) in cultured neurons from AD mice, resulting in decreased Aβ peptide levels, decreased Aβ42/40 ratio, and increased cell survival [66]. Importantly, further studies in vivo indicated that lentiviral injection of dCas9-Dnmt3a in mouse brain induces efficient DNA methylation editing, decreases the levels of APP, and improves the cognitive defects associated with this AD model.

#### 2.1.4. Mitochondrial DNA Methylation

Finally, in addition to nuclear DNA, mitochondrial DNA (mtDNA) methylation has also been observed in humans and animal models. In ALS, *SOD1* mutation carriers display increased levels of mtDNA and demethylation of the mitochondrial D-loop, a noncoding region critical for both mtDNA replication and transcription. This could represent an attempt to compensate for the disease-associated loss of mitochondrial function by mtDNA upregulation in carriers of ALS-linked *SOD1* mutations [67]. While D-loop is also hypomethylated in the hippocampus of a mouse model of AD (APP/PS1), this is associated with decreased mtDNA copy number [68]. Thus, while altered methylation of mtDNA suggests a possible role for this epigenetic mechanism in ALS and AD, the fact that the outcome in the level of mtDNA is opposite in the two cases warrants further studies on the topic.

**Table 1 ijms-23-12609-t001:** Global and gene-specific DNA methylation in prion-like diseases.

Disease	Species/Model	Tissue Type	Methylation Finding	References
Alzheimer’s disease	Human	Brain, peripheral blood	Methylation profiles of AD patients. Identification of differentially methylated positions (DMPs)	[19,20,30,31]
	Human	Brain	Methylation profile of AD patients. Identification of differentially methylated regions (DMRs)	[27]
	Human	Superior temporal gyrus	Hypermethylated DMRs	[28,29]
	APP/PS1 mice	Hippocampus	Changes in mitochondrial DNA methylation	[68]
	Human	Neurons	Hypomethylated enhancers in *DSCAML1* gene	[41]
	Human	Hippocampus, cerebellum, peripheral blood	Hypomethylation in *TOMM40* and *APOE* gene promoters	[61]
	Human	Frontal cortex	Methylation of *PM20D1* gene	[42]
	Human	Peripheral blood	Hypomethylation of *B3GALT4* and *ZADH2* genes	[57]
	Human	Peripheral blood	Hypomethylation of *BIN1* gene	[58]
	Human	Peripheral blood	Hypermethylation of *ERα* gene promoter	[59]
Parkinson’s disease	Human	Peripheral blood	Identification of DMRs between PD patients and healthy controls	[22]
	Human	Brain	Identification of DMRs in PD-affected brain areas	[32]
	Human	iPSC-derived dopaminergic neurons	Global DNA hypermethylation changes	[36,37]
	Human	Peripheral blood, saliva	Global DNA hypomethylation changes	[21,23]
	Human	Peripheral blood, iPSC-derived dopaminergic neurons	Hypomethylation of *SNCA* gene promoter. Reversion of disease symptoms via CRISPR/Cas9-mediated *SNCA* hypermethylation	[43,65]
	Human	Leukocytes	Hypomethylation of *NPAS2* and *DRD2* genes	[60,63]
Amyotrophic lateral sclerosis	Human	Peripheral blood	Methylation profile of ALS patients. Identification of DMPs	[24]
	Human	Brain	Methylation profile of ALS patients. Identification of DMRs	[33]
	Human	Peripheral blood	Increased global 5-methylcytosines levels in sALS and FALS	[53]
	Human	Peripheral blood	Hypermethylation of the *C9orf72* promoter and association of DNA methylation age-acceleration with disease duration and age of onset in *C9orf72* expansion carriers	[54,55]
	Human	Peripheral blood	Increase in global DNA methylation and demethylation of the mitochondrial D-loop region in *SOD1* mutation carriers	[56,67]
	Human	hESCs, iPSCs	hESCs unmethylated and iPSCs hypermethylated at the *C9orf72* repeats	[64]
	Human	Brain, motor neurons	Differential methylation of *KIAA1147*, *IGHMBP2*, *COL15A1*, *TARDBP*, *RANGAP1*, *IGHMBP2*, *OGG1*, *APEX1*, *PNKP*, and *APTX*	[44,45,46]
Huntington’s disease	YAC128 mice	Brain	Methylation profile. Identification of DMRs	[34]
	Human	Peripheral blood	Methylation profile of HD patients. Identification of DMPs	[26]
	Human	Brain	Minimal association of DNA methylation with HD status	[39]
	Human	Peripheral blood	No significant changes between patients and controls	[25]
	Human	Brain	11 co-methylation modules associated with HD status	[35]
	Human	Brain	*HES4* promoter hypermethylation	[47]
	Human	Plasma, saliva	*BDNF* promoter hypermethylation	[48]

### 2.2. Histone Modifications

The most studied histone modification in prion-like diseases is acetylation. Global levels of acetylated histones and differential expression of the enzymes in charge of their acetylation or deacetylation have been related to the pathology of these diseases. Moreover, pharmacological modulation of these enzymes could ultimately result in a potential treatment strategy. Table 2 summarizes the main results in this subject.

Widespread acetylome variation has been observed in different brain areas of AD patients, and this epigenetic modification can be induced by pathological tau [69,70]. Acetylation of a lysine residue in histone 3 (H3K27) varies in the vicinity of several known AD risk genes (*APP*, *CR1*, *MAPT*, *PSEN1*, *PSEN2*, and *TOMM40*) and is robustly associated with the disease [69]. Similarly, altered histone acetylation has been linked to PD-associated neurodegeneration. PD neurotoxins specifically increase histone acetylation through an autophagy-mediated reduction of histone deacetylases (HDACs) in PD patients’ dopaminergic neurons [71].

The modulation of epigenetic enzymes involved in acetylation, mainly HDACs, also has a role in the development and progression of prion-like diseases. HDACs can be divided into four subtypes, classes I, II, and IV being classical HDACs and class III consisting of NAD^+^-dependent silent information regulator 2 family members (sirtuins). HDAC activities are unbalanced in fibroblasts from PD patients, which is associated with impaired mitophagy and increased cell death [72].

In addition, changes in expression levels of particular HDACs have been described, but their modifications are not always in accordance with studies. Modifications can be different depending on the disease and CNS cell populations. HDAC1 and HDAC2 levels are strongly decreased in the frontal cortex of AD patients, and HDAC1 is also reduced in the hippocampus [73]. HDAC2 downregulation seems to contribute to cholinergic nucleus basalis of Meynert neuronal dysfunction, neurofibrillary tangles pathology, and cognitive decline during the clinical progression of AD [74]. On the contrary, HDAC2 is upregulated in the microglia from the substantia nigra of PD patients [75]. Additionally, nuclear accumulation of HDAC4, which is normally localized at the cytoplasm, seems to promote neuronal apoptosis in PD-affected dopaminergic neurons [76].

The levels of class II HDACs (4, 5, and 6) are increased in the skeletal muscle of *SOD1*-ALS mice with severe neuromuscular impairment [77]. Although this increase was not observed in motor neurons, Class II HDACs could also contribute to motor neuron degeneration as their pharmacological inhibition is able to restore the expression and function of glutamate transporter EAAT2 in the spinal cord of ALS [78]. However, HDAC4 expression is decreased in the skeletal muscle of ALS patients [79], and the skeletal muscle-specific ablation of HDAC4 is sufficient to induce an earlier onset of the disease, a decrease in neuromuscular junctions’ size, and muscle denervation and atrophy in *SOD1*-ALS mice [80]. Finally, an in vivo brain assessment of HDAC alterations by positron emission tomography showed no significant differences in HDAC expression levels between ALS patients and healthy controls [81], suggesting that HDAC alterations may have a more profound effect on the disease in peripheral tissues. Given that the skeletal muscle function is compromised in ALS, the implication that HDAC4 could be protective in ALS skeletal muscle has questioned the use of broad-range HDAC inhibitors as a strategy for ALS treatment [82]. However, research on the role of HDACs in different tissues warrants further investigation because the discovery of highly selective inhibitors for HDACs could eliminate the potential negative effects of more broad-range targeting in the future.

In AD patient-derived neurons and triple transgenic (3xTg-AD, carrying mutated forms of *APP*, *PSEN1*, and *MAPT*/tau) mouse model, inhibition of HDAC3 (Class I) decreases pathological tau phosphorylation and acetylation, reduces Aβ protein expression, increases Aβ degradation, improves learning and memory and normalizes several AD-related genes [83]. Importantly, HDAC6 seems to influence tau phosphorylation, autophagic flux, and tubulin acetylation [84], and its inhibition stimulates pathological tau degradation in ADLP APT mice (carrying six mutations in *APP*, *PSEN1*, and *MAPT*/tau) and AD patient-derived brain organoids [85]. The inhibition of HDAC6 has also been proposed as a therapeutic strategy for PD, as this enzyme contributes to oxidative injury and dopaminergic neurotoxicity through mediating deacetylation of peroxiredoxins 1 and 2 in oxidopamine-induced PD mice [86]. In the PD mouse model expressing mutated *LRRK2* (R1441G), HDAC inhibition by valproic acid has a neuroprotective effect through modulation of neuroinflammation and improvement of PD-like symptoms [87]. Inhibition of histone sirtuin-2 deacetylase (SIRT2) also has therapeutic effects protecting degenerating dopaminergic neurons, reducing microglia activation, and facilitating the trafficking and clearance of misfolded proteins [88,89].

This therapeutic approach of HDAC inhibition has also been investigated in ALS models. HDAC inhibition in spinal cord-dorsal root ganglion cultures enables the heat shock response, which manages a load of aberrant proteins in a stress-dependent manner in cultured spinal motor neurons and also rescues the DNA repair response in iPSC-derived motor neurons carrying the *FUS* mutation [90]. Similarly, global histone hypoacetylation was observed in a *FUS* murine model. The restoration of histone acetylation levels in these mice by HDAC inhibition ameliorated the disease phenotype and significantly extended their lifespan [91]. In addition, HDAC6 inhibition restored axonal transport defects and mitochondrial and endoplasmic reticulum vesicle transport defects in ALS patient-derived motor neurons by increasing the α-tubulin acetylation level [92,93].

In HD rats and mouse models, HDAC inhibition produces a variety of neuroprotective beneficial effects, including partial reversal of behavioral symptoms, reversion of aberrant neuronal differentiation [94], prevention of striatal neuronal atrophy, improvement of motor performance [95], amelioration of disease phenotypes in a transgenerational manner [96] and reestablishment of pyruvate dehydrogenase activity improving mitochondrial function and bioenergetics [97]. Moreover, a multiomic study has proposed that the positive effect of HDAC4 knockdown in rescuing synaptic function in HD mice could be a consequence of synaptic vesicle trafficking regulation, and HDAC4 could interact with htt via association with htt-interacting proteins [98]. Other HDACs have also been proposed as therapeutic targets in HD. Striatal HDAC2 levels are reduced in the YAC128 HD mouse model subjected to dietary restriction, which could contribute to improving the disease phenotype [99]. In addition, inhibition of HDAC3 improves motor deficits, suppresses striatal CAG repeat expansions, and reduces the accumulation of oligomeric forms of mutant htt in HD transgenic mice [100,101]. In contrast, genetic deletion of HDAC6 exacerbates social impairments and hypolocomotion in HD R6/1 mice [102].

Although there are fewer works analyzing this epigenetic mechanism compared to DNA methylation or microRNAs, the results obtained seem to be promising, not as a source of biomarkers but as a possible target for therapies.

Other histone modifications have been studied, but the number of these studies is even lower. A genome-wide study in human brains has identified differential enrichment of trimethylated lysine 4 of histone 3 (H3K4me3) mark between HD and control samples [103]. Interestingly, in a Drosophila melanogaster HD model, the activity reduction of the H3K27-specific demethylase, Utx, ameliorated neurodegeneration and diminished htt aggregation [104]. Conversely, in this same model, histone methyltransferase dSETDB1/ESET was identified as a mediator of mutant htt-induced degeneration [105].

**Table 2 ijms-23-12609-t002:** Histone post-translational modifications in prion-like diseases.

Disease	Species/Model	Tissue Type	Main Finding	References
Alzheimer’s disease	Human	Brain	Widespread acetylomic variation associated with AD, possibly induced by pathological tau	[69,70]
	Human	Frontal cortex, hippocampus	Decreased levels of HDAC1	[73]
	Human	Frontal cortex	Decreased levels of HDAC2 contributing to neuronal dysfunction, neurofibrillary tangles pathology, and cognitive decline	[73,74]
	3xTg-AD mice, Human	Human iPSC-derived neurons	HDAC3 inhibition decreases pathological tau phosphorylation and acetylation	[75]
	ADLP^APT^ mice, Human	Human-derived brain organoids	HDAC6 inhibition stimulates pathological tau degradation	[77]
Parkinson’s disease	Human	Dopaminergic neurons	PD neurotoxins increase histone acetylation through an autophagy-mediated HDACs reduction mechanism	[71]
	Human	Fibroblasts	Imbalance between total HATs and HDACs activities	[72]
	LRRK2 R1441G mice	Brain	HDAC inhibition has a neuroprotective effect through modulation of neuroinflammation and improvement of PD-like behaviors	[79]
	Sirt2 −/− C57-BL6 mice, Human	Brain	Sirtuin-2 deacetylase inhibition protects degenerating dopaminergic neurons, reduces microglial activation, and facilitates the trafficking and clearance of misfolded proteins	[80,81]
	Human	Microglia from the substantia nigra	Upregulation of HDAC2	[83]
	E13-14 mice	Dopaminergic neurons	HDAC4 accumulation promotes neuronal apoptosis	[84]
	C57-BL6 mice	Brain	HDAC6 could contribute to oxidative injury	[86]
Amyotrophic lateral sclerosis	*SOD1*-ALS mice	Skeletal muscle	Increase in class II HDACs (4, 5, and 6) involved in modulating the expression and function of glutamate transporter	[85,86]
	Human	iPSC-derived motor neurons	HDAC inhibition rescues the DNA repair response	[90]
	Tg *FUS* +/+ mice	Spinal cord	Global histone hypoacetylation. Restoration of histone acetylation ameliorates the disease phenotype	[91]
	Human	iPSC-derived motor neurons	HDAC6 inhibition restores axonal transport defects and mitochondrial and endoplasmic reticulum vesicle transport defects	[92,93]
	Human	Brain	No significant differences in HDAC expression levels between patients and controls	[97]
	*SOD1*-ALS mice, Human	Skeletal muscle	Decreased expression of HDAC4 associated with an earlier onset of the disease	[87,88]
Huntington’s disease	tgHD rats, BACHD, R6/2, YAC128 mice	Brain	HDAC inhibition produces neuroprotective beneficial effects	[94,95,96,97]
	HttQ20, HttQ140 mice	Brain	HDAC4 regulates synaptic vesicle trafficking and interacts with htt	[98]
	YAC128 mice	Brain	Reduction of HDAC2 could contribute to improving the disease phenotype	[99]
	N171-82Q, HdhQ^111^ knock-in mice	Brain	HDAC3 inhibition improves motor deficits, suppresses striatal CAG repeat expansions, and reduces accumulation of mutant huntingtin oligomeric forms	[100,101]
	R6/1 mice	In vivo assessment	HDAC6 deletion exacerbates social impairments and hypolocomotion	[102]
	Human	Brain	Differentially enrichment of H3K4me3 mark	[103]
	*Drosophila melanogaster*	Brain, eye	Activity reduction of Utx ameliorates neurodegeneration and diminishes htt aggregation. dSETDB1/ESET might be a mediator of mutant htt-induced degeneration	[104,105]

### 2.3. MicroRNAs

A large number of human proteins containing prion-like domains are RNA or DNA-binding proteins [106]. The first protein with these motifs associated with neurodegeneration was TDP-43 (trans-activation response element DNA-binding protein 43) [107,108], which is the major pathological protein in sporadic ALS. This protein plays a role in many RNA-related functions, including microRNA biogenesis. Afterward, many other proteins related to neurodegenerative diseases have been discovered that play a role in RNA metabolism, and amyloids coaggregate with endogenous nucleic acids [109,110]. Although different RNA species may be altered in prion-like diseases, we focus on the role of microRNA in these pathologies. These biomolecules have been widely studied in neurodegenerative diseases. Changes in miRNA expression profiles may serve as biomarkers, and these molecules seem to play important roles in the neuropathology of these diseases.

#### 2.3.1. MicroRNA Profiles

Several studies have measured miRNA expression profiles in tissues and body fluids from patients and in different in vitro and animal models of prion-like diseases. Sets of microRNAs altered in each prion-like disease seem to be largely different, as summarized in Table 3.

In AD patients, miRNA profiles have been analyzed in brain [111,112,113,114], blood [115,116], serum [117,118], CSF [119] and extracellular vesicles (EVs) [120,121,122]. Brain and EVs miRNA profiles have as well been characterized in AD mouse models [123,124,125]. Comparing the different profiles, only one miRNA, miR-324-3p, is commonly downregulated in the brain, body fluids, and EVs of AD patients, but its expression is not altered in AD mouse models. MiRNA profiles have been also characterized in brain [126,127,128,129], gut [130], blood [128], plasma [131,132], serum [133], CSF [134,135], saliva [136], and EVs [131,137] of PD patients; in brain of a PD mouse model [135]; in blood of a PD rat model [138]; and in a PD in vitro model [139]. Although no common miRNAs are found when comparing all the above-mentioned PD profiles, there are common miRNAs between some tissues and body fluids. MicroRNA miR-451a is upregulated in the brain, gut, and CSF of PD patients and also in the brains of PD mice, and miR-19b-3p is downregulated in the brain, blood, plasma, and saliva of PD patients.

In ALS patients, microRNAs have been studied in brain [140,141], skeletal muscle [142,143], blood [143,144,145,146], plasma [147], serum [148,149,150], CSF [151] and EVs [152,153,154,155]. Distinctive ALS miRNA profiles are also present in mouse skeletal muscle [156] and serum [157] and in in vitro models [154,158]. Three miRNAs, miR-125a-3p, miR-193a-5p, and miR-455-3p, are commonly downregulated in the brain, skeletal muscle, blood, and serum from ALS patients, and the expression of miR-125a-5p, which is downregulated in the brain and skeletal muscle of ALS patients, is also reduced in ALS mouse models and in vitro models. A smaller number of studies have explored the miRNA profiles in HD. These profiles are found in the brain [159,160,161], plasma [162], and CSF [163] of HD patients and in the brain of HD mouse models [164,165,166,167]. No common miRNAs are found between the profiles performed in HD patients, but there are two miRNAs, miR-10b-5p and miR-10a-5p, that are upregulated in the brain of both HD patients and mouse models.

When comparing the human miRNA profiles of the four prion-like diseases, there are two miRNAs, miR-144-3p and miR-22-5p, that appear upregulated in all four pathologies (Figure 2a). The β-amyloid precursor protein (APP) has been identified as a target gene of miR-144-3p, which seems to have a role in mitochondrial function maintenance [168], and a potential neuroprotective role has been predicted for miR-22 related to the regulation of targets implicated in HD such as HDAC4 [169]. Further analyses are necessary to verify if these two microRNAs display important roles in misfolded protein-related diseases.

Conversely, although there are miRNAs similarly downregulated between two or three prion-like diseases, no common downregulated miRNAs are found between the four diseases (Figure 2b). The existence of differently regulated microRNA in different diseases can be used as a source of biomarkers for differential diagnosis. Appendix A lists common upregulated and downregulated miRNAs for combinations of different prion-like diseases and the datasets used to generate the Venn diagrams.

#### 2.3.2. The Role of microRNAs in Prion-Like Diseases

In addition to miRNA profiles that could reveal potential disease biomarkers, some studies have shed light on the functions or possible roles of different miRNAs in prion-like diseases.

In AD pathology, several miRNAs seem to have neuroprotective roles attenuating Aβ accumulation and its associated neurotoxicity. In particular, the following miRNAs have been associated with this neuroprotective function: miR-193a-3p [170] and miR-133b [171] in human serum, miR-335-5p [172] and miR-361-3p [173] in the human brain, miR-200a-3p [174] in plasma from AD patients, miR-107 [175] in human neuroblast cell lines, and miR-340 [176] in the hippocampus from senescence-accelerated (SAMP8) mice that show increased oxidative damage associated with APP overproduction. Moreover, in rat hippocampus, miR-134-5p [177] is involved in rescuing AD synaptic plasticity deficit, and miR-124 [178] and miR-200a-3p [174] are involved in alleviating tau pathology in murine and in vitro models, respectively. On the other hand, miR-34c [179] and miR-124 [180] mediate synaptic and memory deficits in AD. Other miRNAs recently described to likely participate in AD risk and progression are miR-146a, miR-181a, detected in blood from AD patients, and miR-142-3p [181,182] in the human brain. All aforementioned miRNAs have been proposed as potential diagnostic biomarkers and/or therapeutic targets for AD pathology.

Dysregulation of miRNAs is also implicated in PD pathogenesis. The upregulation of several miRNAs, including miR-150 [183] in human serum, miR-let-7a [184], and miR-190 [185] in C57BL/6 mouse model and miR-135b [186] in in vitro models, ameliorates PD-associated neuroinflammation. By inhibiting SP1, a transcription factor expressed in the brain, miR-375 decreases dopaminergic neurons’ damage, reduces oxidative stress, and diminishes inflammation in PD, and miR-29c also attenuates dopaminergic neuron loss, neuroinflammatory response, and α-synuclein accumulation [187,188]. Furthermore, in a PD rat model, miR-3557 and miR-324 seem to be involved in delaying PD neurodegeneration [189], and overexpression of miR-410 in a PD cellular model appears to exert neuroprotective effects against apoptosis and reactive oxygen species production [190]. In contrast, overexpression of miR-326 has been described to promote autophagy of dopaminergic neurons, and miR-195 downregulation might induce microglia-mediated neuroinflammation activation [191,192]. On the other hand, miR-7 seems to be involved in regulating *BDNF* expression in the early stages of PD, and miR-376a could also be implicated in PD pathogenesis, possibly by regulating the expression of mitochondrial-related genes [193,194].

Several studies also report miRNA changes in ALS, suggesting that these molecules could play a role in the development and progression of the disease. In serum from ALS patients, downregulation of miR-335-5p may enhance mitophagy, autophagy, and apoptosis pathways [195]. Interestingly, in the spinal cord of an ALS mouse model, downregulated miR-375-3p appears to control various target structures that intervene at different sites of the apoptosis pathway [196]. In contrast, in the cerebellum of an ALS mouse model, increased miR-29b-3p seems to downregulate proapoptotic factors, leading to neuroprotection [197]. Regarding to axon degeneration, several miRNAs, namely miR-126-5p [198], miR-494-3p [199] and miR-1825 [200], might facilitate this pathological feature in ALS. On the other hand, upregulated miR-338-3p is responsible for decreased glycogenolysis and subsequent glycogen accumulation within the spinal cord of *SOD1*-ALS mice [201], and miR-105 and miR-9 seem to potentially contribute to the pathogenesis of intermediate filament inclusions in ALS [202]. Furthermore, it has been described that extracellular miR-218 released from dying motor neurons in ALS can be taken up by neighboring astrocytes and negatively affect astrocyte function [203]. In the skeletal muscle of FALS patients, it has been reported an upregulation of miR-206, involved in the neuromuscular junction, regeneration, and muscle atrophy, and also an increase in inflammatory miRNAs (miR-27a, miR-221, miR-155) [79]. miR-206, a microRNA that was consistently altered during the course of the disease in the skeletal muscle of the *SOD1*-G93A ALS mouse model, is also increased in serum from ALS patients [155]. Mechanistically, it has been suggested that increased miR-206 is an attempt to promote maintenance and/or repair of neuromuscular junctions by targeting and inhibiting HDAC4, which leads to a fibroblast growth factor (FGF)-stimulated reinnervation [204]. Finally, reduced expression of the miR-17~92 cluster has been associated with the vulnerability of limb-innervating lateral motor column motor neurons to ALS-related degeneration [205].

In HD, a consistent association between expression profiles of CSF-miRNAs and the earliest prodromal stages of the disease has been reported [162]. In contrast, a study in a knock-in mouse model of HD (Hdh mice) suggests that miRNA regulation may have a limited global role in responding to HD in the striatum and cortex of these mice [206]. A decrease in miR-132 has been observed in the brain of another model of HD (HD R6/2 mice), and restoration of miR-132 deficiency seems to confer amelioration in motor function and lifespan of these mice [207]. In neural progenitors and differentiated neural cells of a transgenic HD nonhuman primate model and in HD murine primary neurons, miR-196a has shown neuroprotective effects, including improvement of cell survival and mitochondrial functions, reduction of cytotoxicity and apoptosis and enhancement of neuronal morphology and differentiation [208,209]. Regarding htt, miR-27a has been reported to reduce mutant htt aggregation in R6/2-derived neuronal stem cells [210]. Interestingly, artificial miRNAs have also been used to successfully reduce mutant htt levels in a transgenic HD sheep model and in a humanized Hu128/21 HD mouse model [211,212].

As reported, the role of microRNAs in these neurodegenerative diseases is better known than the ones of the other epigenetic mechanisms. We have the tools to analyze the effect of overexpressing or repressing the expression of these molecules using expression vectors or antisense oligonucleotides in cellular models. This could facilitate the research in therapies modifying dysregulated microRNAs.

**Table 3 ijms-23-12609-t003:** MicroRNAs in prion-like diseases.

Disease	Species/Model	Tissue Type	miRNA	Main Finding	References
Alzheimer’s disease	Human	Brain, peripheral blood, serum, CSF, serum and CSF exosomes, and plasma extracellular vesicles	miRNA profiles	Differential miRNA expression profiles in AD patients	[111,112,113,114,115,116,117,118,119,120,121,122]
	APP/PS1 and 5XFAD mice	Brain and urinary exosomes	miRNA profiles	Differential miRNA expression profiles in AD mouse models	[123,124,125]
	Human	Serum	miR-193a-3p, miR-133b	Neuroprotective roles attenuating Aβ accumulation and its associated neurotoxicity	[170,171]
	Human	Brain	miR-335-5p, miR-361-3p		[172,173]
	SAMP8 mice	Hippocampus	miR-340		[175]
	Human	SH-SY5Y, SK-N-SH cells	miR-107		[176]
	Human	Blood plasma	miR-200a-3p		[174]
	Rat	Hippocampus	miR-134-5p	Involved in rescuing AD synaptic plasticity	[177]
	C57BL/6J, Tg2576 mice	Hippocampus	miR-124	Alleviates tau pathology and mediates synaptic and memory deficits	[178,180]
	SAMP8 mice, Human	Hippocampus, serum	miR-34c	Mediates synaptic and memory deficits	[179]
	Human	Blood, Brain	miR-146a, miR-181a, miR-142-3p	Associated with AD risk and progression	[181,182]
Parkinson’s disease	Human	Brain, gut, plasma, serum, CSF, saliva, plasma exosomes, serum extracellular vesicles, and iPSC-derived dopaminergic neurons	miRNA profiles	Differential miRNA expression profiles in PD patients	[126,127,128,129,130,131,132,133,134,135,136,137]
					[139]
	Rat	Peripheral blood	miRNA profile	Differential miRNA expression profile in a PD rat model	[138]
	Human	Serum	miR-150	Amelioration of PD-associated neuroinflammation	[183]
	C57BL/6 mice	Brain	miR-let-7a, miR-190		[184,185]
	Human, rat	SH-SY5Y, PC-12 cells	miR-135b		[186]
	Wistar rats, C57BL/6 mice, Human	Brain, SH-SY5Y cells	miR-375, miR-29c	Decrease in dopaminergic neurons’ damage and neuroinflammatory response	[187,188]
	Sprague–Dawley rats	Brain	miR-3557, miR-324	Involved in delaying PD neurodegeneration	[189]
	Human, rat	SH-SY5Y, PC-12 cells	miR-410	Overexpression exerts neuroprotective effects against apoptosis and reactive oxygen species production	[190]
	C57BL/6 mice	Brain	miR-326	Overexpression promotes autophagy of dopaminergic neurons	[191]
	Mouse	BV2 cells	miR-195	Downregulation might induce microglia-mediated neuroinflammation activation	[192]
	Sprague–Dawley rats	Peripheral blood, brain	miR-7	Involved in regulating brain-derived neurotrophic factor expression in early stages of PD	[193]
	Human	PBMCs, SH-SY5Y cells	miR-376a	Implicated in PD pathogenesis regulating the expression of mitochondrial-related genes	[194]
Amyotrophic lateral sclerosis	Human	Brain, spinal cord, skeletal muscle, neuromuscular junctions, plasma, serum, leukocytes, CSF, plasma, serum, brain and spinal cord extracellular vesicles, motor neuron-derived exosomes, iPSC-derived motor neurons, and motor neuron progenitors	miRNA profiles	Differential miRNA expression profiles in ALS patients	[140,141,142,143,144,145,146,147,148,149,150,151,152,153,154,155]
					[158]
	SOD1^G86R^ and SOD1^G93A^ mice	Serum, skeletal muscle	miRNA profiles	Differential miRNA expression profiles in ALS mouse models	[156,157]
	Human	Serum	miR-335-5p	Downregulation may enhance mitophagy, autophagy, and apoptosis pathways	[195]
	Wobbler mice	Spinal cord	miR-375-3p	Regulates target structures that intervene at the apoptosis pathway	[196]
	Wobbler mice	Cerebellum	miR-29b-3p	Downregulates proapoptotic factors, leading to neuroprotection	[197]
	SOD1^G93A^ mice	Muscle	miR-126-5p	Involved in axon degeneration	[198]
	Human	Astrocytes	miR-494-3p		[199]
	Human	CNS	miR-1825		[200]
	SOD1 mice	Spinal cord	miR-338-3p	Upregulation decreases glycogenolysis causing glycogen accumulation within the spinal cord	[201]
	Human	Spinal cord	miR-105, miR-9	Contribute to intermediate filament aggregation in ALS	[202]
	Mouse	Astrocytes	miR-218	Affects astrocyte function negatively	[203]
	Human, SOD1^G93A^ mice	Skeletal muscle, serum, plasma	miR-206	Upregulated. Involved in neuromuscular junction, regeneration, and muscle atrophy	[79,156]
	Human	Skeletal muscle	miR-27a, miR-221, miR-155	Increased in FALS patients	[79]
	Human, SOD1^G93A^ mice	Motor neurons	miR-17~92 cluster	Associated with vulnerability of motor neurons to ALS-related degeneration	[205]
Huntington’s disease	C57BL/6, R6/1 and BACHD mice	Brain	miRNA profiles	Differential miRNA expression profiles in HD mouse models	[164,165,166,167]
	Human	Brain, plasma, CSF	miRNA profiles	Differential miRNA expression profiles in HD patients. Association between CSF-miRNAs expression profiles and the earliest prodromal stages of HD	[172,173,174,175,176]
	Hdh mice	Striatum, cortex	miRNA profiles	miRNA regulation may have a limited global role in responding to HD	[206]
	R6/2 mice	Brain	miR-132	Decreased levels, whose restoration confers amelioration in motor function and lifespan	[207]
	FVB mouse embryos, HD1/HD7/WT monkey	Neural progenitors, neural cells	miR-196a	Neuroprotective effects	[208,209]
	R6/2 mice	Neuronal stem cells	miR-27a	Reduces mutant htt aggregation	[210]
	Hu128/21 mice, sheep	Striatum	Artificial miRNAs	Reduce mutant htt levels	[211,212]

## 3. Epigenetic Changes in Prion Diseases

Compared to prion-like diseases, there are very few studies on the involvement of epigenetic changes in transmissible spongiform encephalopathies.

It is known that the prion protein (PrP) is able to bind to RNA and DNA molecules [213]. As these nucleic acids can induce PrP aggregation, they have been proposed as catalysts in the conversion of the PrP^C^ to the pathologic form PrP^Sc^ [214]. Different DNA molecules are capable of binding to recombinant PrP (rPrP), resulting in complex aggregates [215]. Interestingly, the GC content of these DNA molecules seems to be important in the binding, affinity, stability, and aggregation abilities and in the toxic species generation [215]. A recent study has evaluated the neurotoxic effect of the inoculation of a PrP-DNA complex in the lateral ventricle of Swiss mice, which, after inoculation, showed cognitive impairment, hippocampal synapse loss, and intense glial activation [216]. In contrast, in human neuroblastoma cell cultures, PrP cytotoxicity is attenuated when combined with DNA molecules, which stabilize PrP structure and reduce its pathogenic properties [217].

Different RNA molecules can also bind to PrP and trigger its aggregation and conversion to PrP^Sc^, the efficiency of the conversion depending on the RNA source [216,218,219]. Some of these RNA molecules, specifically RNA aptamers, are able to bind and stabilize PrP^C^ and reduce PrP^Sc^ levels in infected mouse neuronal cells [220]. Further studies are needed in order to evaluate the involvement of these nucleic acids in prion pathology and to develop potential therapeutic strategies.

### 3.1. DNA Methylation Profiles

DNA methylation might also be involved in the pathogenesis of prion diseases. A study of the mouse prion protein gene (*PRNP*) encoding the PrP^C^ protein has reported an association between DNA methylation and *PRNP* gene expression. The *PRNP* gene promoter region seems to be unmethylated, and the methylation status of one of the *PRNP* enhancer regions was negatively correlated with *PRNP* expression [221]. In addition, during neuronal differentiation of mouse embryonic carcinoma P19C6 cells, the expression of *PRNP* was markedly increased, while CpG methylation was significantly reduced, suggesting that DNA methylation could be implicated in mediating *PRNP* expression regulation [222].

Only three studies have analyzed the methylome of prion diseases. A genome-wide methylation study has shown differentially methylated positions in blood from sporadic Creutzfeldt–Jakob disease (sCJD) patients compared with healthy controls, some of these positions correlated with disease progression [223]. Furthermore, another genome-wide methylation study of the CNS of sheep naturally infected with scrapie has identified differentially methylated regions between control and scrapie animals, belonging some of them to genes with possible neuroprotective roles and to genes that may contribute to scrapie disease progression [224]. Interestingly, a recent study was able to identify different DNA methylation patterns in tonsil and appendix lymphoreticular tissues between sCJD patients and healthy individuals [225], pointing to a potential source of diagnostic biomarkers in prion diseases.

Although these three works have been performed in different tissues (blood, CNS, and lymphoid tissues) and species (CJD human patients and scrapie-infected sheep) and have used different methodologies (450k methylation array for humans and whole genome bisulfite sequencing), we have compared the genes detected in each work containing either DMR or DMP. Twelve genes were differentially methylated in both ovine CNS and CJD blood. Further studies are necessary to confirm the role of these common genes in prion diseases. Appendix A shows the common DMR and DMP in prion and prion-like diseases.

We have compared the methylation DMR and DMP profiles between these studies and those performed in prion-like diseases. As different methodology has been used, only DMPs obtained in CJD samples were compared with the other diseases. Only 2 DMPs identified in CJD blood were found in common with AD, 3 with PD, and 1 with ALS. However, of a total of 8907 DMRs observed in CNS of scrapie-infected sheep, 634 are also altered in PD, 60 in Alzheimer’s, 51 in ALS, and 61 in HD. This large difference is due to the fact that PD and scrapie were analyzed using the same methodology, bisulfite-treated DNA sequencing. More such studies are needed in the other prion-like diseases to determine whether or not there are common genes differentially methylated in all prion-misfolded pathologies.

### 3.2. Histone Modifications

Yeast ESI+, for expressed sub-telomeric information, is the prion form of the Set3C histone deacetylase scaffold Snt1 (NCOR1 in humans). This prion, in response to cell cycle arrest, is able to activate gene expression through H4 acetylation and RNA polymerase II recruitment [226]. Other histone deacetylases, namely HDAC6 and sirtuin-1 (SIRT1), have shown protective effects during prion infection. Although many studies in prion-like diseases are addressed to inhibit HDAC6 as a potential therapeutic approach, in cerebral cortical neurons, overexpression of HDAC6 alleviates prion peptide-mediated neuronal cell death and toxicity [227]. On the other hand, overexpression of SIRT1, which is decreased in the brains of scrapie-infected rodents and in prion-infected SMB-S15 cells, reduces PrP^Sc^ levels and protects against prion protein-induced neuronal cell death and mitochondrial dysfunction [228,229,230]. Therefore, although HDAC inhibition is proposed as a potential therapeutic strategy in prion-like diseases, this approach does not seem the best choice for treating prion diseases. Further analysis will be necessary to find out the role of other HDACs in prion diseases and to design novel therapies that maybe could be focused on the overexpression of HDACs.

### 3.3. MicroRNA Profiles

A number of studies have reported changes in miRNA expression profiles during prion infection in the CNS, plasma, and synaptoneurosomes of preclinical and clinical scrapie-infected mice [231,232,233], in the serum of elk infected with the chronic wasting disease (CWD) [234] and in plasma of naturally infected classical scrapie sheep [235]. When comparing these profiles with the ones reported in prion-like diseases, there are some miRNAs commonly altered in the two groups of diseases. Table 4 shows the common upregulated, and downregulated miRNAs, and Appendix A lists the datasets generated from the different miRNA profile studies in prion diseases. The prion-like disease with more miRNAs in common with prion diseases is ALS, whereas the one with fewer miRNAs in common is HD. More studies are required in order to find specific miRNAs that are only altered in prion diseases, even only in each type of prion disease, specifically for their use as diagnostic biomarkers.

Although the potential role of most of these alterations in TSE pathology is unknown, multiple miRNAs regulate PrP^C^ levels both directly and indirectly in human neuroectodermal cell lines [236]. On the other hand, miR-16, which is increased in hippocampal neurons during presymptomatic prion disease, could decrease neurite length and branching, probably via the downregulation of components of the MAPK/ERK pathway [237]. Additionally, a single nucleotide polymorphism in miR-146a has been associated with susceptibility to FFI (fatal familial insomnia) and with the appearance of some clinical features in sCJD patients [238]. Interestingly, artificial miRNAs have also been used to reduce PrP^C^ and subsequently suppress PrP^Sc^ propagation in primary mixed neuronal and glial cells culture [239].

## 4. Concluding Remarks

It is evident that epigenetic mechanisms, namely DNA methylation, histone post-translational modifications, and microRNAs, are involved in the pathophysiology of neurodegenerative diseases. Increasing evidence in prion-like diseases and, more recently, in prion diseases has shown that epigenetic modifications can modulate different pathogenic mechanisms occurring in these neurodegenerative disorders. Nevertheless, these epigenetic mechanisms seem to act differently in each of these diseases. Global DNA methylation changes have been detected in prion diseases and prion-like diseases in a variety of tissues and in several specific genes, showing different trends in the global methylation profile of each disease and several genes harboring differentially methylated regions and positions that match between some of these diseases. Modulation of HDAC enzymes seems to also be a common epigenetic mechanism in these diseases. Interestingly, among all the HDACs, the enzyme HDAC6 is involved in both prion diseases and prion-like diseases, although the mechanism of action is different between the two groups of diseases. On the other hand, other HDCAs, namely HDAC2, HDAC3, and HDAC4, are associated with different aspects of prion-like diseases. Additionally, miRNA profiles seem to be the most specific of each disease. However, there are two miRNAs, miR-144-3p and miR-22-5p, that seem to be commonly upregulated in the four prion-like diseases but not in prion diseases, and other miRNAs, such as miR-335-5p, miR-375, and miR-27a, have functions in AD, PD, HD, and ALS, but they seem to regulate different pathways in each disease. Prion and prion-like diseases also share several miRNAs in common, ALS being the prion-like disease with the highest number of common miRNAs. In addition, artificial miRNAs have successfully been used in both HD and prion diseases to reduce misfolded protein levels. Further research is still needed, especially in prion diseases where knowledge is still scarce, in order to elucidate the exact molecular pathways by which these epigenetic mechanisms perform their regulatory roles and to identify potential epigenetic diagnostic and therapeutic biomarkers.

## Figures and Tables

**Figure 1 ijms-23-12609-f001:**
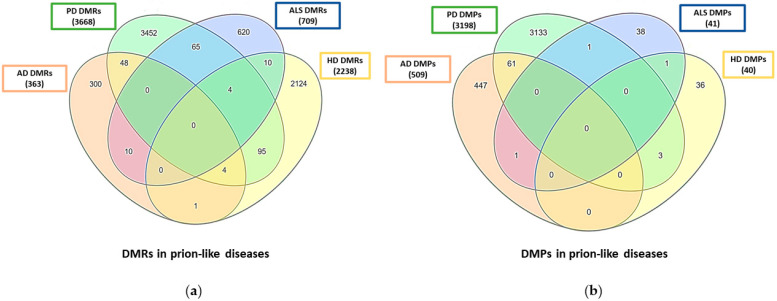
Comparison of differentially methylated regions (DMRs) and positions (DMPs) in prion-like diseases. Venn diagrams indicate the number of common and unique DMRs (**a**) and DMPs (**b**) in AD, PD, ALS, and HD.

**Figure 2 ijms-23-12609-f002:**
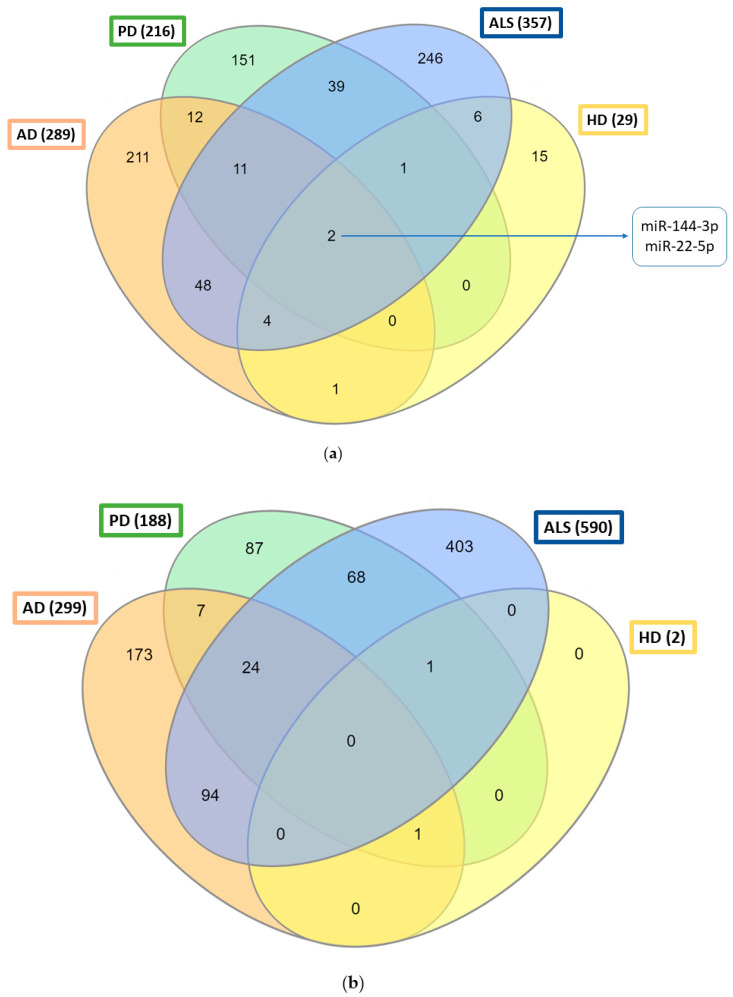
Venn diagrams of miRNA profiles in prion-like diseases: (**a**) upregulated miRNAs in prion-like disease patients; (**b**) downregulated miRNAs in patients suffering from prion-like diseases.

**Table 4 ijms-23-12609-t004:** MicroRNAs commonly upregulated and downregulated in prion and prion-like diseases. (+) = Upregulated and (−) = Downregulated.

miRNA	Prion Diseases	AD	PD	ALS	HD	Upregulated (+)/Downregulated (−)
miR-5100	+		+	+		Upregulated
miR-342-3p	+/−	−	+/−	+/−		Upregulated/Downregulated
let-7f-5p	+		+	+		Upregulated
miR-146b-5p	+	+	+	+		Upregulated
let-7a-5p	+	+		+		Upregulated
miR-378c	+	+		+		Upregulated
miR-27a-3p	+	+		+		Upregulated
miR-339-3p	+	+		+		Upregulated
miR-142-5p	+	+	+	+		Upregulated
miR-146a-5p	+	+				Upregulated
miR-320a-3p	+/−	+	+	−		Upregulated/Downregulated
miR-10a-5p	+				+	Upregulated
miR-326	+			+		Upregulated
miR-21-5p	+			+		Upregulated
miR-324-5p	+			+		Upregulated
miR-103a-3p	+			+		Upregulated
miR-331-3p	+/−		−	+/−		Upregulated/Downregulated
miR-107	+			+		Upregulated
miR-142-3p	+			+		Upregulated
miR-129-5p	−		−	−		Downregulated
miR-423-5p	−		−	−		Downregulated
miR-125a-5p	−		−	−		Downregulated
miR-148a-3p	−		−	−		Downregulated
miR-186-5p	−		−	−		Downregulated
miR-141-3p	−		−	−		Downregulated
miR-149-5p	−			−		Downregulated
miR-200a-3p	−			−		Downregulated
miR-200b	−			−		Downregulated
miR-323-3p	−			−		Downregulated
miR-338-3p	−	−		−		Downregulated
miR-342-5p	−			−		Downregulated
miR-382	−			−		Downregulated
miR-383	−			−		Downregulated
miR-433	−			−		Downregulated
miR-455-5p	−			−		Downregulated
let-7b	−			−		Downregulated
let-7c	−			−		Downregulated
miR-486-3p	−			−		Downregulated
miR-183-5p	−			−		Downregulated
miR-100-5p	−			−		Downregulated
miR-125b-5p	−			−		Downregulated
miR-99a-5p	−			−		Downregulated
miR-145-3p	−			−		Downregulated
miR-410-3p	−			−		Downregulated
miR-181d-5p	−			−		Downregulated
miR-375	−			−		Downregulated
miR-99b-5p	−			−		Downregulated
miR-30e-3p	−	−	−	−		Downregulated
miR-129-2-3p	−	−	−	−		Downregulated
miR-223-3p	−	−	−	−		Downregulated
miR-493-3p	−	−				Downregulated
miR-182-3p	−	−				Downregulated
miR-877-5p	−	−		−		Downregulated
miR-182-5p	−	−		−		Downregulated
miR-144-5p	−	−		−		Downregulated
miR-181c-5p	−	−		−		Downregulated

## Data Availability

Data sharing is not applicable to this article.

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
