# Peer review of "Epigenetic Changes in Prion and Prion-like Neurodegenerative Diseases: Recent Advances, Potential as Biomarkers, and Future Perspectives"

_ijms, 2022, doi:10.3390/ijms232012609_

Round 1
Reviewer 1 Report
Overall this review represents an impressive amount of work. Thé review is mainly descriptive with no or litte thoughts or hypothesis on the origin and nature of the pathogenic agent, In fact the authors state since the beginning that the pathological agent corresponds to misfolded protein molecules, named prion protein or Prp that become insoluble upon agregration. Such protein aggregates are endowed with a high level OF stability achieved by multiple interactions of Prp with the NA backbone, phosphate residues and bases, notably G. Prp aggregates, either fibrils or plaques can be exported in the extracellular space causing cell stress, damages and death. How are the aggregates formed? The most recent data srongly suggest that Prp, especially when in excess, bind small RNA molecules leading to the formation of rather complex RNP's where Prp can recruit enzymes such as DNApolymerases, nucleotidyl transferases, RNA ligases, and methylases In addition such aggregates were found go contain fair amounts of small RNA molecules in accordance with the fact that PrP is an RNA binding protein endowed with potent RNA chaperoning activities. Thus it is entirely possible the nature of the pathoogic prion agent be an RNP capable of reprogramming several cellular reactions, such as NAreplication and translation . This is not discussed in the presnet review, nor is the possibility that Prp is a Janus chaperone driving the conformation of nucleic acids and of proteins attthe same time, and thus of RNP ensembles. Along this line Prp can form stable complexes with enzymes such as RT and IN of retroviral or cellular origin, resulting in an extensive activation of RT and IN activities and fidelity. Also not discussed at all in the present review is another characteristics of Prp that can protect small RNAs such as siRNA against degradation by cellular and viral nucleases Another critique concerns the section on HDAC which appears highly summarized wile at the same time autoirs do not consider RNA modifications, especially at the RNA 5' end to which Prp provides protection. SInce the RNA 5' end is multifunctional impacting on RNA stabilty, translation, turnover and kick starting formation of RNP ensembles, the interplay of Prp with the RNA5' end is of primary importance and could well be a hall mark of prion diseases. Not discussed at all in this review.` Taken together this review ms cannot be published in its present form.;; In a possible revised version authors should take into account the most recent findings on Prp and its possible roles in RNA metabolism, as well as its roles in viral infections (most probably a restriction factor)Author Response
We would like to thank this reviewer for his/her suggestions. The aim of the review was to identify common epigenetic mechanisms (DNA methylation, histone modification and microRNAs) in prion-like neurodegenerative diseases and their possible translation to prion diseases, where the knowledge is still scarce. Nevertheless, we have included in the manuscript some paragraphs describing prions as possible epigenetic mechanisms and their role in RNA metabolism.
- Introduction: page 2, lines 81-88
- Chapter 2.3 MicroRNAs: page 10, lines 1146-1155
- Chapter 3 Epigenetic changes in prion diseases: page 17, lines 1352-1368 and page 18 lines 1412-1415
Reviewer 2 Report
Article ID: IJMS-1929452
Title: Epigenetic changes as biomarkers of prion diseases and prion-like neurodegenerative diseases: recent advances and future perspectives
Authors: Hernaiz A; Toivonen JM; Bolea R; Martin-Burriel I
Summary: In this manuscript, the authors review the roles of epigenetic changes, including methylation, histone modification, and microRNAs in the pathogenesis of prion and prion-like disorders. The authors review what is known in each category as it relates primarily to Alzheimer’s disease, Parkinson’s disease, Huntington’s disease, and ALS, with some additional comments on the dearth of knowledge on the subject as it relates to prion disorders. The discussion of therapeutic targets and the use of various biomarkers is helpful to put the complex basic science studies and their results into context. As the manuscript is a review, my comments will focus primarily on organization rather than content.
Comments: This is a well-written manuscript that summarizes a large amount of data, sometimes conflicting, on epigenetic studies involving a wide range of prion-like disorders. The authors note that there is not much information available for the classic prion disorders (i.e., CJD, BSE, etc.). Based on that, the authors may wish to address two minor concerns I have prior to final acceptance of this manuscript.
Minor concerns:
1) The title of the manuscript suggests there is an equivalent amount of data available between prion and prion-like disorders. The authors may wish to re-evaluate their title to consider the scarcity of information available on classic prion disorders.
2) Organizationally, there is a lot of conflicting information, primarily between different prion-like disorders. The authors do a good job of breaking these down and explaining them but there are a lot of “however” and “on the other hand” statements. This repeated language might have a tendency to get readers spun around in circles. The authors may wish to modify these statements somehow to provide more clarity. The concepts of overlapping and contrasting data is inherent in the Venn diagrams provided, but the syntax of several paragraphs could use a bit of work (the paragraph covering Lines 144-167 especially, but here and there throughout the manuscript as well).
All in all, I think this is a good review on the subject of epigenetics and prion/prion-like disorders, though the authors may wish to reinforce through the title the scarcity of information on prion disorders specifically. I would like to thank the authors and the editor for the opportunity to review this manuscript, and I look forward to any revisions with updated data the authors may have in the future.
Author Response
We would like to thank this reviewer for the effort he/she made reading the manuscript and the suggestions made. You can find below how we have responded to each of the comments made by this referee.
"Minor concerns:
1) The title of the manuscript suggests there is an equivalent amount of data available between prion and prion-like disorders. The authors may wish to re-evaluate their title to consider the scarcity of information available on classic prion disorders."
- We are aware that the amount of data is scarce in prion diseases. Moreover, prion-like diseases include ALS, Alzheimer’s, Parkinson’s and Hungtington’s diseases. The aim of the review was to compare prion-like diseases trying to find common epigenetic mechanisms to translate them to prion research, where fewer studies have been performed. We have slightly modified the title and have mentioned in the text that the information in prion diseases is limited.
- Introduction: Page 1, lines 90-91
- Concluding remarks: Page 21, lines 1496-14972
"2) Organizationally, there is a lot of conflicting information, primarily between different prion-like disorders. The authors do a good job of breaking these down and explaining them but there are a lot of “however” and “on the other hand” statements. This repeated language might have a tendency to get readers spun around in circles. The authors may wish to modify these statements somehow to provide more clarity. The concepts of overlapping and contrasting data is inherent in the Venn diagrams provided, but the syntax of several paragraphs could use a bit of work (the paragraph covering Lines 144-167 especially, but here and there throughout the manuscript as well)."
- Syntax has been revised along the text trying to make the manuscript easier to read and clearer. As the manuscript has been modified using track changes, this reviewer can see the corrections we made.
"All in all, I think this is a good review on the subject of epigenetics and prion/prion-like disorders, though the authors may wish to reinforce through the title the scarcity of information on prion disorders specifically. I would like to thank the authors and the editor for the opportunity to review this manuscript, and I look forward to any revisions with updated data the authors may have in the future. "
- We would like to thank this reviewer for his/her positive comments and the suggestions made. It is complicated include this issue in a short title so we have not modified it. As we said above, the scarcity of works in prion diseases is stated in abstract (page 1, lines 20-21), introduction and concluding remarks.
Reviewer 3 Report
In this review, the authors aim to discuss the possible roles of different types of epigenetic modifications (DNA methylation, Histone modifications and miRNA profiles) in the pathogenesis of prion diseases (TSE) and prion-like neurodegenerative diseases (ND). This is an important topic which is likely to appeal to a large audience of scientists. In fact, I think each type of epigenetic modification discussed in this manuscript and its impact on ND could be te focus of its own review.
Having said that, I found the review to be too descriptive and linear in its organization. No clear take-home message or useful information emerges after reading it (are epigenetic modifications a cause or a consequence? Are they useful as diagnostic or therapeutic tools? What are the mechanisms involved? What type of investigations are needed in the future?).
The authors list different, often contradictory, studies suggesting possible links between epigenetic modifications and prion-like diseases, but without providing a critical view of all these studies (which are most to trust? What are the models and experimental approaches used and what are their limitations? etc.). The authors should also put their own work into perspective; how does their views fit within the large amount of data published on these complex issues.
The title is misleading: 1) there is no clear discussion about how epigenetic changes can be used as BIOMARKERS; 2) FUTURE PERSPECTIVES are not discussed in depth.
The authors state that they used InteractiVenn do generate the Venn diagrams in Figure 1-4; but with which datasets? What were the selection criteria? The datasets do not appear comparable for each disease (e.g. only 363 DMR for AD but 3668 for PD); this weakens the conclusions made by the authors regarding the number of different/common DMRs and DMPs.
In Figure 1 and 2, the data presented in the pie charts can already be found in the Venn diagrams; there is no need for both representations. The text in light yellow or green is hard to read.
Tables 4 and 5 should be combined by adding an upregulated/dowreguated column.
Author Response
We would like to thank this reviewer for the effort made in reading and revising the manuscript and for the suggestions proposed, which we believe have contributed to its improvement. '
Below we detail point by point how we have responded to their comments.
“Having said that, I found the review to be too descriptive and linear in its organization. No clear take-home message or useful information emerges after reading it (are epigenetic modifications a cause or a consequence? Are they useful as diagnostic or therapeutic tools? What are the mechanisms involved? What type of investigations are needed in the future?)."
- The authors list different, often contradictory, studies suggesting possible links between epigenetic modifications and prion-like diseases, but without providing a critical view of all these studies (which are most to trust? What are the models and experimental approaches used and what are their limitations? etc.). The authors should also put their own work into perspective; how does their views fit within the large amount of data published on these complex issues.”
- We have divided in sub-chapter the different parts of the manuscript and tried to get conclusions of the bibliographic research we have made. The reviewer can see along the text all the discussion and extra-information we have added, including the necessity of further studies. We hope this new version of the manuscript was more clear and interesting for your and future readers.
- We have also discussed the work we and others found in prion diseases.
“The title is misleading: 1) there is no clear discussion about how epigenetic changes can be used as BIOMARKERS; 2) FUTURE PERSPECTIVES are not discussed in depth.”
- Following the comments of the reviewer, the revised manuscript includes more information about biomarkers and future perspectives. We have slightly modified the title but left the role of biomarkers and future perspectives. If, after reading the manuscript, this reviewer agrees with the changes made, we will leave the title as is.
“The authors state that they used InteractiVenn do generate the Venn diagrams in Figure 1-4; but with which datasets? What were the selection criteria? The datasets do not appear comparable for each disease (e.g. only 363 DMR for AD but 3668 for PD); this weakens the conclusions made by the authors regarding the number of different/common DMRs and DMPs.”
- Supplementary Table S1 lists the datasets used to generate the Venn diagrams and the common genes that contain DMRs or DMPs for all combinations of different prion-like diseases (page 3 lines 158-159, page 21 line 1502 ).
- The datasets contain the data from all available works. We have not applied any selection criteria. However, we have included a fragment explaining the limitations of this comparison concerning the tissue used, the number of studies performed and the different techniques used for the different diseases (Page 3, lines 164-176).
“In Figure 1 and 2, the data presented in the pie charts can already be found in the Venn diagrams; there is no need for both representations. The text in light yellow or green is hard to read.”
- We have removed pie charts, left just Figure 1 and changed the colour of characters to black.
“Tables 4 and 5 should be combined by adding an upregulated/dowreguated column.”
- We combined the tables 4 and 5.
Round 2
Reviewer 3 Report
I would like to thank the authors for their efforts into addressing the issues I have with the first version of their review.The manuscript is now very much improved, although still a bit descriptive and linear in its organization (but this is common with this type of reviews). In my opinion, the review is acceptable for publication in its current form.